# Genome-Wide Identification of the Long Noncoding RNAs of *Tribolium castaneum* in Response to Terpinen-4-ol Fumigation

**DOI:** 10.3390/insects13030283

**Published:** 2022-03-14

**Authors:** Hailong Wu, Shuaili Yue, Yong Huang, Xinping Zhao, Haiqun Cao, Min Liao

**Affiliations:** 1Anhui Province Key Laboratory of Crop Integrated Pest Management, School of Plant Protection, Anhui Agricultural University, Hefei 230036, China; whl6340@163.com (H.W.); yueshuaili153@163.com (S.Y.); yongh2016@163.com (Y.H.); zxinping1014@163.com (X.Z.); haiquncao@163.com (H.C.); 2Anhui Province Engineering Laboratory for Green Pesticide Development and Application, School of Plant Protection, Anhui Agricultural University, Hefei 230036, China

**Keywords:** lncRNA, *Tribolium* *castaneum*, terpinen-4-ol, metabolism, RNA-seq

## Abstract

**Simple Summary:**

Long noncoding RNAs (lncRNAs) are important regulatory factors in multiple biological processes, including genomic imprinting, cancer, RNA interference, and protein translation. Several lncRNAs can respond to insecticides. However, lncRNA functions associated with terpinen-4-ol resistance in the red flour beetle (*Tribolium castaneum*) have not yet been identified. In previous work, we found terpinen-4-ol to have strong fumigation activity against store-product pests. As a pesticide from plants, terpinen-4-ol shows nearly no residual danger to the environment; however, resistance is inevitable if people use terpinen-4-ol immoderately. To avoid resistance to terpinen-4-ol occurring in the red flour beetle, we deeply sequenced and tried to find some lncRNAs that can regulate target mRNA expression to reduce terpinen-4-ol.

**Abstract:**

Long noncoding RNAs (lncRNAs) are important regulatory factors in multiple biological processes, and several lncRNAs are known to respond to insecticides. However, the lncRNA functions that are associated with terpinen-4-ol resistance in the red flour beetle (*Tribolium castaneum*) have not yet been identified. In this study, we determined the differentially transcribed lncRNAs between fumigated and control experimental groups. In the six libraries that underwent RNA sequencing, 34,546 transcripts were identified, including 8267 novel lncRNAs, 4155 novel mRNAs, 1151 known lncRNAs, and 20,973 known mRNAs. Among these, we found that the expression of 1858 mRNAs and 1663 lncRNAs was significantly different in the fumigated group compared with the control group. Among the differentially transcribed lncRNAs, 453 were up-regulated and 1210 were down-regulated lncRNAs. In addition, we identified the regulatory function targets of the lncRNAs. Functionally, all lncRNAs and target genes associated with terpinen-4-ol metabolism were enriched in several metabolic pathways, like the ATP-binding cassette transporter, pentose interconversion, and glucuronate interconversion. To the best of our knowledge, this study represents the first global identification of lncRNAs and their potential association with terpinen-4-ol metabolism in the red flour beetle. These results will provide reference information for studies on the resistance to terpinen-4-ol and other essential oil compounds and chemical pesticides, as well as an understanding of other biological processes in *T. castaneum*.

## 1. Introduction

The red flour beetle, *Tribolium castaneum*, is a major worldwide pest of stored products, particularly food grains, and is a model organism for food safety and genome research [1,2]. Insects and diseases cause 25% of the loss of stored grain worldwide every year [3], and controlling stored-product pests has become increasingly necessary. However, *T. castaneum* currently demonstrates resistance to control due to the reckless overuse of traditional pesticides [4,5].

Terpinen-4-ol is the primary component of *Melaleuca alternifolia* (tea tree) oil, and it has been shown to have many biological activities. These activities include suppression of human inflammatory mediator production [6], inhibition of the growth of human melanoma cells in vitro [7], and control of dimorphic fungi [8] and pathogens [9]. In addition, tea tree oil has high acaricidal [10,11] and pesticide activities [12]. Terpinen-4-ol and other components of the essential oil (EO) showed high activity as fumigants in controlling stored-product pests and cockroaches [13,14,15,16], suggesting the potential prospect of developing a fumigation pesticide based on tea tree oil. To date, few articles have reported on *T. castaneum* strains that have developed resistance to terpinen-4-ol.

Resistance to pesticides such as cyfluthrin, pyrethroids, carbaryl, deltamethrin, and cyclodiene is associated with an increased glutathione (GSH) concentration and glutathione S-transferase (GST) activity, microsomal oxidation, high expression of cytochrome P450 monooxygenase, and single point mutations in resistance to the dieldrin gene (*Rdl*) [5,17,18,19].

Some differentially expressed genes enriched in metabolism by P450, GST, UDP-glucuronosyltransferase (UGT), and ABC transporters were identified in our previous study, which were associated with the fumigation activity of terpinen-4-ol [20,21]. Knockdown of *cyp6ms1* and the NADPH-cytochrome P450 reductase (*CPR*) gene showed that *cyp6ms1* mediated the susceptibility of *S. zeamais* to terpinen-4-ol by feeding wheat mixed with dsRNA [12].

LncRNAs are defined as non-protein-coding transcripts longer than 200 nucleotides that are transcribed by RNA polymerase II, similar to mRNAs. The different categories of lncRNA functions (sense, antisense, intronic, and intergenic) are mainly related to genomic imprinting, posttranscriptional control, and epigenetic processes [22,23]. Accumulating studies have shown that lncRNAs have critical functions in human diseases such as cancer, but their functions in insects are only just beginning to be investigated.

RNA-seq is a powerful method for identifying lncRNAs and has revealed lncRNAs in multiple insect species, including *Helicoverpa armigera* [24], *Nilaparvata lugens* [25], *Bombyx mori* [26], *Aedes aegypti* [27,28], *Drosophila melanogaster* [29], *Aphis gossypii* [30], and *Zeugodacus cucurbitae* [31], following different treatments in *Plutella xylostella* [24,32,33,34,35] and in stages of *T. castaneum* [36]. Moreover, lncRNAs have been characterized in several of the species mentioned above in association with the immune response [37]. A previous study reduced the expression of a lncRNA in intron 20 of cadherin alleles and found the decreased transcription of *PgCad1* and decreased susceptibility to Cry1Ac in *Pectinophora gossypiella* [38]. Nevertheless, no studies have focused on understanding the resistance of pesticides in *T. castaneum*, especially regarding terpinen-4-ol. Once lncRNAs associated with terpinen-4-ol stress were identified, pri-miRNAs and target mRNAs could be predicted, and the lncRNA-miRNA-mRNA regulation pathway of reducing terpinen-4-ol activity can be found in *T. castaneum*. In this study, we assembled the RNA transcripts of *T. castaneum* before and after terpinen-4-ol treatment and identified the lncRNAs related to mRNA by reducing terpinen-4-ol activity. These lncRNAs may serve as gene regulators and miRNA precursors.

## 2. Materials and Methods

### 2.1. Insect Culture and Treatment

The *Tribolium*
*castaneum* insects were reared in our laboratory without any pesticide exposure. The larvae were fed whole-wheat flour (with 5% yeast) and kept at 28 ± 2 °C with 70% relative humidity (RH) in complete darkness in the climate chamber. Eggs were collected every 2 days and laid in a new bottle to make sure the pests were all at the same developmental stage. Seven days after eclosion, adults were fumigated with terpinen-4-ol (Acros, Belgium) at LC_50_ for 24 h; control insects were not exposed. Triplicate groups were used for the treatment and control.

### 2.2. RNA Extraction, Library Establishment, and Sequencing

Ten adults were collected from each group into an Eppendorf tube as a combined sample to extract total RNA. TRIzol reagent (Invitrogen, Carlsbad, CA, USA) was used to extract total RNA according to the manufacturer’s procedure. Then, the extracted RNA underwent 1% agarose gel electrophoresis for assessment of RNA integrity and quality, and DS-11 (Denovix, Wilmington, DE, USA) was used to measure RNA concentration.

Six libraries (triplicates of the treatment and control group) were established, and RNA-seq was performed by BGI Genomics (Shenzhen, China). A Ribo-Zero^TM^ rRNA Removal Kit (Illumina, San Diego, CA, USA) was used to remove rRNA from total RNA, a TruSeq^®^ Stranded Kit (Illumina, San Diego, CA, USA) was used to synthesize first strand-specific cDNA, and DNA polymerase and RNaseH were used to synthesize double-strand cDNA. The final cDNA libraries were purified by polymerase chain reaction (PCR) of the ligation products of double-stranded cDNA. Libraries were sequenced on the Illumina HiSeq 2500 system. A flow chart can be seen in the Appendix A.

### 2.3. lncRNA Bioinformatic Analysis

The raw reads were cleaned by filtering out rRNA and low-quality reads containing poly N bases. SOAP [39] was used to align reads to the Ribosomal Database Project (http://rdp.cme.msu.edu/) (accessed on 7 September 2019), and matched sequences were removed, resulting in sequences called clean reads. HISAT [40] was used to align the clean read into the *T. castaneum* genome (GCA_000002335.3) which was assembled by StringTie [41]. Transcripts were compared with mRNAs and lncRNAs to obtain their location information using Cufflinks [42]. Cufflinks also assembled integrated transcripts. All transcripts were quantitatively, differentially, and cluster and enrichment analyzed. We also predicted the targeted genes of the identified lncRNAs, the lncRNAs’ family, and the potential pri-miRNAs (Appendix A).

### 2.4. Coding Capacity and Express Analysis of lncRNA

The coding capacity of the transcripts was calculated by the Coding Potential Calculator (CPC) [43], txCdsPredict, the Coding-Noncoding Index (CNCI) [44], and the Pfam database [45]. Transcripts revealing a coding potential with a CPC score > 0, a txCdsPredict score > 500, a CNCIscore > 0, and a Pfam-scan > 0.001 were considered to be mRNAs; otherwise, they were considered to be lncRNAs. The transcript was identified as mRNA or lncRNA only when the judgments were consistent among at least three of these methods.

Clean reads were aligned to the reference sequence by Bowtie2 [46], and RSEM [47] was used to calculate the expression of genes and transcripts. To compare expression levels between samples, FPKM was used to normalize gene expression levels. The FPKM method helped eliminate the influence of different gene lengths and sequencing quantities on the calculated gene expression. Therefore, the calculated gene expression quantity can be used directly to compare the gene expression levels between different samples.

### 2.5. Differential Gene Expression Analysis

Differential gene expression was determined using the DEGseq R package [48]. A fold change ≥ 2 and adjusted *p*-value ≤ 0.001 defined significant differential expression.

### 2.6. Functional Annotation and Gene Enrichment Analysis

Blast [49] was used to annotate the KEGG pathway and Blast2GO [50] was used to annotate the GO terms for lncRNAs and targeted mRNAs. A false discovery rate ≤ 0.01 and *p*-value ≤ 0.05 were considered indicative for significantly enriched GO terms and KEGG pathways.

### 2.7. Prediction Targeted Genes and Families of lncRNAs

The function of lncRNAs is principally realized by *cis*- and *trans*-patterns to the targeted gene. The basic principle of *cis*-targeted gene prediction is that the function of lncRNA is related to the protein-coding genes adjacent to its coordinates; therefore, a *cis*-pattern lncRNA was considered to be within 10 kb of the upstream or 20 kb of the downstream mRNA. Beyond this range, RNAplex [51] was used to calculate the binding energy between the lncRNA and the mRNA. If the binding energy was less than or equal to 30, a *trans*-pattern lncRNA was identified. Spearman and Pearson’s coefficients were calculated first, with Spearman_cor ≥ 0.6 and Pearson_cor ≥ 0.6. By alignment with the Rfam database [52], lncRNAs were divided into different noncoding RNA families using NFERNAL [53].

### 2.8. Real-Time Quantitative PCR Analysis

To confirm the sequencing results, total RNA was extracted from another set of 10 adults from the two treatments, and first-strand cDNA was synthesized (Yeasen, China) and used for validation by RT-qPCR (qPCR SYBR Green Master Mix, Yeasen, China, and performed on Bio-Rad CFX96 Touch, USA) using the following method: 95 °C for 2 min; 45 cycles of 95 °C for 10 s and 60 °C for 30 s; with the machine’s default melting curve used. RPL18 served as the reference gene. All lncRNA primers were obtained from Primer-BLAST, an online primer-designing tool (https://www.ncbi.nlm.nih.gov/tools/primer-blast/) (accessed on 11 December 2020). The results of the validation were calculated by 2^−ΔΔct^ [54], and the graph was generated using GraphPad 6 (GraphPad Software, San Diego, CA, USA) for comparison with the RNA-seq data. Specific primers were designed using Beacon Designer 8 software (Premier Biosoft International, San Francisco, CA, USA) and are detailed in Table 1.

## 3. Results

### 3.1. Identification and Characterization of the lncRNAs in Tribolium castaneum

In total, 820,027,322 raw reads were generated from the six libraries, with an average of 136,671,220 reads. After removal of all low-quality raw reads (containing adapters, too much N, or abundant low-quality base reads), 752,761,524 clean reads were retained. The total mapping ratio and the unique mapping ratio accounted for more than 65%, with most exceeding 70% (Table 2). The raw transcriptome data were deposited to the SRA database, with the BioProject accession ID of PRJNA808823.

### 3.2. Analysis of Coding Capacity and Differentially Transcribed lncRNAs

Protein-coding capacity analysis of the transcripts predicted that the control samples had an average of 7275 novel lncRNAs, 2630 novel mRNAs, 845 known lncRNAs, and 12,324 known mRNAs. The transcripts from the treated samples were predicted to have an average of 7130 novel lncRNAs, 2605 novel mRNAs, 814 known lncRNAs, and 12,276 known mRNAs (Table 3). These samples contained 8344 novel lncRNAs and 4444 novel mRNAs, as predicted by CPC, CNCI, txCdsPredict, and the Pfam database (Figure 1). Among the lncRNAs, characterization of their genomic locations revealed that 97% (9492) had a number of exons ranging from one to four, and 120 (3%) had at least five exons (Figure 2A). A total of 88% (8553) of the lncRNAs had only one transcript; none of them had more than nine transcripts (Figure 2B), and lncRNAs less than 2.5 kb in length accounted for 83% (8123) of the total (Figure 2C). To determine whether the *T. castaneum* lncRNAs had similar features, we measured the overall expression level (FPKM) of the lncRNAs and found that it was significantly lower than that of mRNAs in *T. castaneum* (Figure 2D).

To systematically assess the *T. castaneum* metabolism-associated lncRNAs of terpinen-4-ol, it was critical to identify the differentially transcribed lncRNAs between treated and control samples. We identified 15,318 differentially expressed mRNAs and 8658 differentially transcribed lncRNAs (Appendix A). Among these, 1858 mRNAs and 1663 lncRNAs were significantly differentially expressed, including 1381 known and 477 novel mRNAs and 1539 novel and 124 known lncRNAs (Figure 3A).

Among the significantly differentially transcribed lncRNAs, we identified 453 up-regulated and 1210 down-regulated lncRNA genes (Figure 3B). Almost 20% of the identified lncRNAs were located on chromosome 3 (Figure 3C).

To validate the RNA-seq data, up- and down-regulated lncRNAs from 11 terpinen-4-ol treatment and control groups were randomly selected, and their relative transcription levels were quantified by RT-qPCR (Figure 3D). Original Ct value were firstly calculated and then the standard error of the mean was analyzed by statistical analysis in SPSS 25. The results of the RT-qPCR and RNA-seq were compared, and the same up and down trends were observed, suggesting that the RNA-seq data were trustworthy and reliable.

### 3.3. Targeted and Family Analysis of the Metabolism-Associated lncRNAs

The function of lncRNAs is principally determined by *cis* and *trans* actions with respect to the target gene. In Appendix A, 10 kb of the upstream or 20 kb of the downstream of the associated mRNA are shown. Our results revealed that 3788 lncRNAs affected 845 mRNAs upstream, including 1568 lncRNAs overlapping with 1846 mRNAs, and 4244 lncRNAs affected 4982 mRNAs downstream without any overlaps (Appendix A). In the same chain, five lncRNAs were completely located in the exons of six mRNAs, and 153 lncRNAs were in the introns of 712 mRNAs. On the other chain, 62 and 143 lncRNAs were located in the exons of 74 mRNAs and the introns of 150 mRNAs, respectively (Figure 4A). lncRNAs were aligned into the Rfam database, and annotation detailed the families to which they belonged. LincRNA accounted for 85.9% of the total lncRNAs (Figure 4B), and we found that 16.88% of the lncRNAs that we identified belonged to the following families: SCARNA7, MIR821, CrcZ, LSU_rRNA_eukarya, Histone3, OppA_thermometer, DLEU2_1, psRNA6, TeloSII_ncR43, and alpha_tmRNA (Figure 4C).

### 3.4. Functional Analysis and Enrichment of the Metabolism-Associated lncRNAs and Target Genes

To understand the function of the significantly differentially transcribed or expressed lncRNAs and the targeted genes, we performed GO and KEGG pathway analyses. The lncRNAs were enriched in biological processes (2291), cellular components (3771), and molecular functions (2823). The associated functions included cellular processes, metabolic processes, membranes or membrane parts, cells or cell parts, and binding and catalytic activities (Figure 5A), most of which were down- or up-regulated (Figure 5B). Via the KEGG pathway analysis of the lncRNAs, we found that 21 pathways (Q value ≤ 0.05) were enriched. These included metabolic pathways (618) and metabolism of ascorbate and aldarate (44), caffeine (26), xenobiotics, and drug metabolism by P450s (113) or other enzymes (76), such as GSTs and UPDs. In addition, neuroactive ligand–receptor interaction (137) and ATP-binding cassette (ABC) transporters (82) were also enriched with the identified lncRNAs (Figure 5B).

LncRNAs function by regulating mRNAs, so the targeted gene enrichments and KEGG pathways have the same importance as the lncRNAs. Cellular processes, metabolic processes, membranes or membrane parts, cells or cell parts, and binding and catalytic activities were the primary forms of GO enrichment (Figure 6A). In the KEGG pathways, targeted genes were also found in 21 pathways; however, this differed from the lncRNA results, with the targeted genes heavily converging on the metabolism of glycerophospholipid, porphyrin and chlorophyll, sphingolipid, ascorbate, and aldarate by P450s or other enzymes (Figure 6B).

## 4. Discussion

In recent years, lncRNAs have been identified and characterized in an increasing number of species following the development of RNA-seq technology. Existing studies primarily focus on mammals, especially in relation to human cancer. The study of insect lncRNAs is still in its preliminary phases, with lncRNAs having been identified in some representative agricultural insects, such as *H. armigera* [24], *N. lugens* [25], *Sogatella furcifera* [55], *A. gossypii* [30], *Z. cucurbitae* [31], and *P. xylostella* [24,32,33,34,35].

In this study, a total of 10008 lncRNAs were identified, of which 8344 were novel (Figure 1A) and 1664 were previously identified lncRNAs. A total of 27,041 mRNAs were identified, including 4444 novel (Figure 1B) and 22,597 known mRNAs. Among those, 8658 lncRNAs showed differential transcription, and nearly 20% (1663) of the lncRNAs had significantly differential transcription (Figure 3A). In previous research, dsEGFP induced differential transcription of 474 lncRNAs in *T. castaneum*, 335 in *H. armigera*, and 593 in *P. xylostella* [24]. In another study, *T. castaneum* was found to gain 4516 lncRNAs during five stages of growth [36]. Our study found even higher numbers of lncRNAs, possibly indicating that compared with dsRNA, xenobiotics may cause more lncRNAs to be expressed to regulate target transcript expression. Another study indicated that suppressing PgCad1 lncRNA could significantly reduce *PgCad1* transcription [38].

Terpinen-4-ol is the most effective insecticidal component from *M. alternifolia* oil reported for *S. zeamais* [21]. Simultaneously, previous work showed that terpinen-4-ol could inhibit Na^+^, K^+^-ATPase activity in *Culex pipiens pallens* and *Musca domestica* [56,57]. Other pesticides, including chlorinated hydrocarbon compounds, have also been shown to inhibit Na^+^, K^+^-ATPase activity in cockroaches and honey bees [58]. Therefore, Na⁺/K⁺-ATPase was initially hypothesized to be the target of terpinen-4-ol. However, in this work, RNA-seq data revealed that lncRNAs and target protein-coding genes are enriched in ABC transporters and P450 and other types of enzymatic metabolisms in *T. castaneum*. This finding is consistent with our previous results that indicated that fumigated *S. zeamais* mRNAs were enriched in metabolic pathways [21]. The different results between our study and other studies may due to the different experimental insect species [56,57]. Another possibility is that the effect of terpinen-4-ol fumigation is due to inhibition of Na^+^, K^+^-ATPase activity in insects rather than Na^+^, K^+^-ATPase being the direct receptor of terpinen-4-ol.

Some reports have shown that monoterpenoids have neurotoxic effects in insects [59]. In recent years, EOs have been tested in mammals and insects and found to affect the following receptors: γ-aminobutyric acid (GABA), tyramine, dopamine, octopaminergic acid, 5-HT, and transient receptor potential (TRP) [60,61,62,63,64,65]. In addition, acetylcholinesterase (AChE) was also considered to be one of the receptors targeted by EOs, which tended to guide research and led to AChE being one of the most investigated mechanisms of action for EOs. However, EOs are rather weak inhibitors of AChE [66]. Another hypothesis centers on EOs affecting insect growth regulators as a means of pest control. It was recently reported that juvenile hormone agonists and antagonists were affected by EOs, causing retardation in the ovarian development of female *A. albopictus* [67]. EOs could be potential pesticides with nonenvironmental pollution characteristics, but studies on their actions in insects are still limited.

In conclusion, we identified and characterized lncRNAs in *T. castaneum* by RNA-seq and used the direction of the lncRNAs to explain the EO compound’s mechanism of action; however, further experiments are needed for verification. This study showed that terpinen-4-ol-induced lncRNAs were concentrated among metabolic processes and biological regulations and enriched in metabolic pathways, which indicates that these lncRNAs up-regulate mRNAs to defend against allogenic material stress. These results increase our understanding of the function of the red flour beetle lncRNAs in resisting xenobiotics and provide a theoretical basis for further research on the function of these lincRNAs and mRNAs related to terpinen-4-ol and the development of fumigants for plant essential oils in *T. castaneum* and other insects.

## Figures and Tables

**Figure 1 insects-13-00283-f001:**
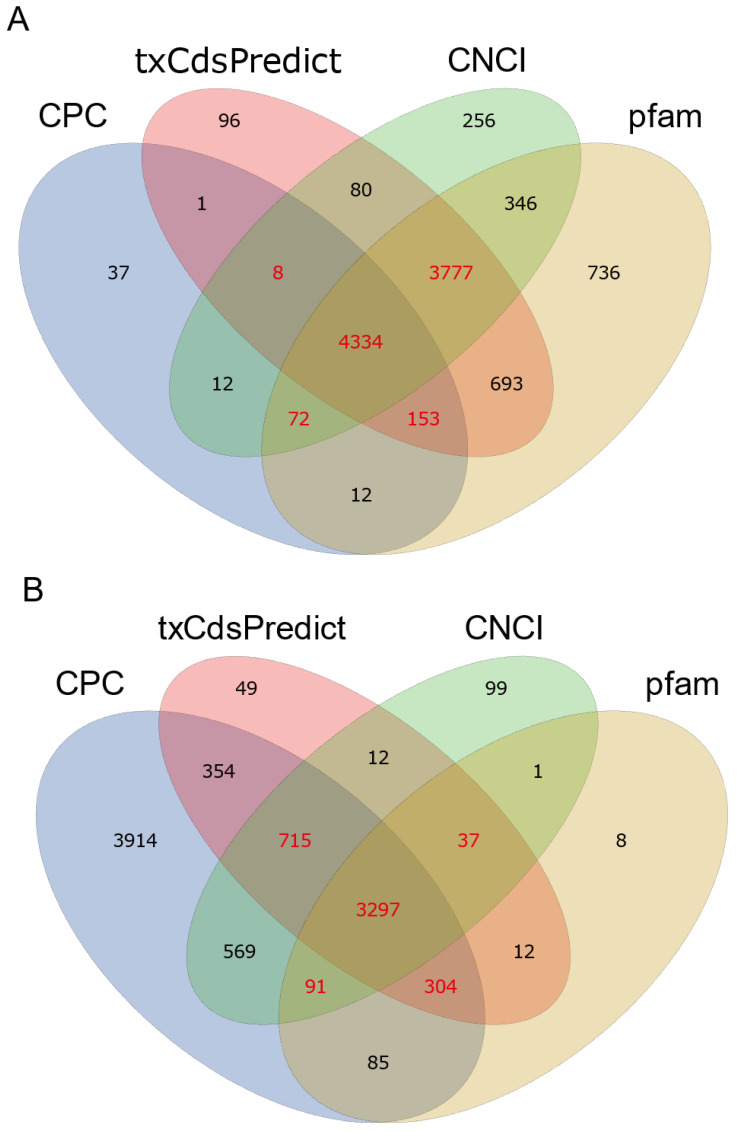
Venn diagram of the novel lncRNAs (**A**) and mRNAs (**B**) by coding capacity analysis using four methods. CPC: Coding Potential Calculator; txCdsPredict: part tool of UCSC Genome Browser; CNCI: Coding-Noncoding Index; pfam: Pfam database. Specific LncRNAs and mRNAs were identified if consistently discovered by at least three of the methods and are shown in red.

**Figure 2 insects-13-00283-f002:**
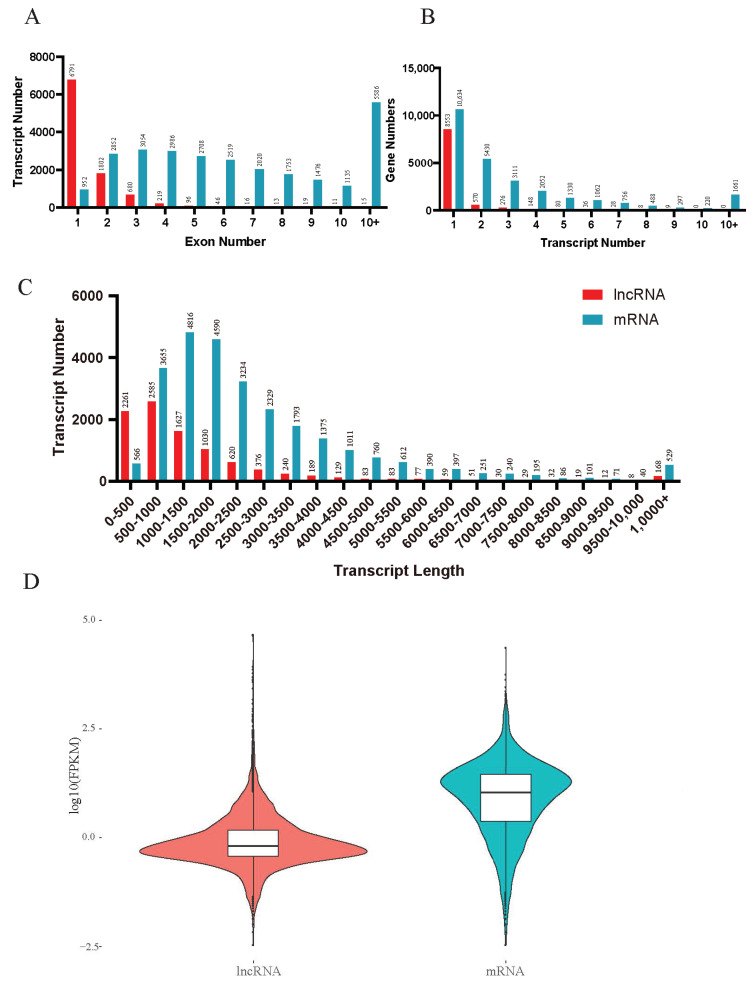
The proportion of exon number (**A**), transcript number (**B**), transcript length (**C**), and the violin diagram were used to compare the FPKM values of lncRNAs and mRNA (**D**). lncRNAs are shown in red, while light blue represents mRNAs.

**Figure 3 insects-13-00283-f003:**
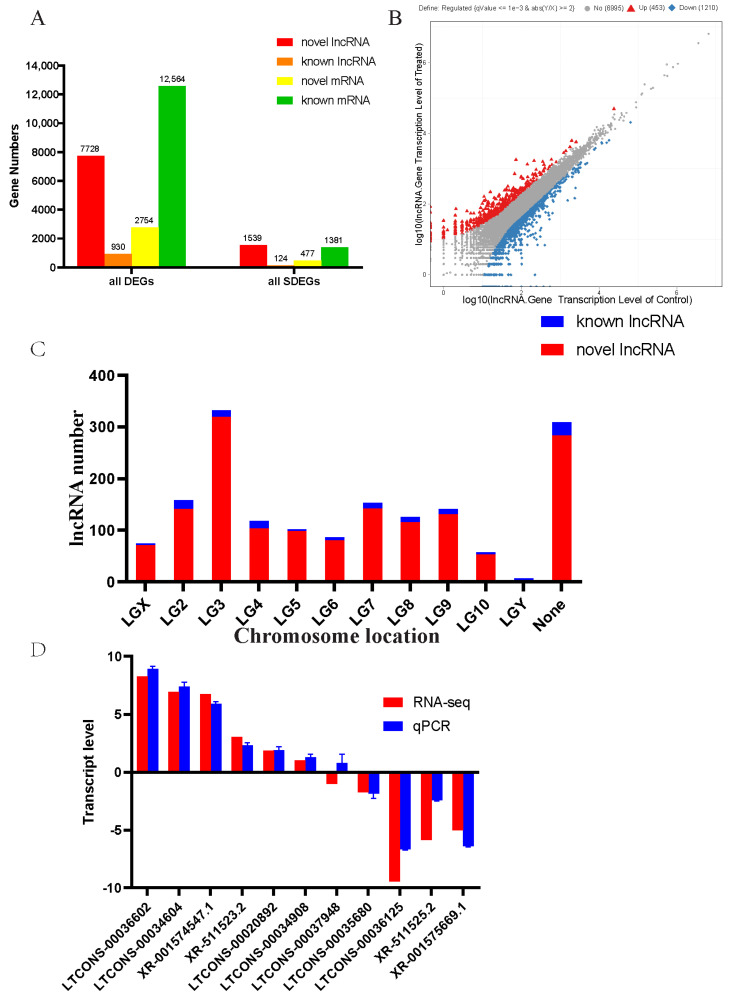
Numbers of identified differentially transcribed or expressed lncRNAs and mRNAs (**A**). (**B**) Scatter diagram showing differentially transcribed lncRNAs between the control and terpinen-4-ol treatment groups, where DEGs represent differentially transcribed or expressed lncRNAs and mRNA, and all SDEGs indicate significantly differentially transcribed or expressed lncRNAs and mRNA. (**C**) Chromosomal distribution of differentially transcribed lncRNAs, where the X-axis indicates the chromosome location of lncRNAs. None represents lncRNAs that matched with unplaced scaffolds. (**D**) Relative transcription of the lncRNAs validated RNA-seq data with qPCR, the X-axis indicates the lncRNAs.

**Figure 4 insects-13-00283-f004:**
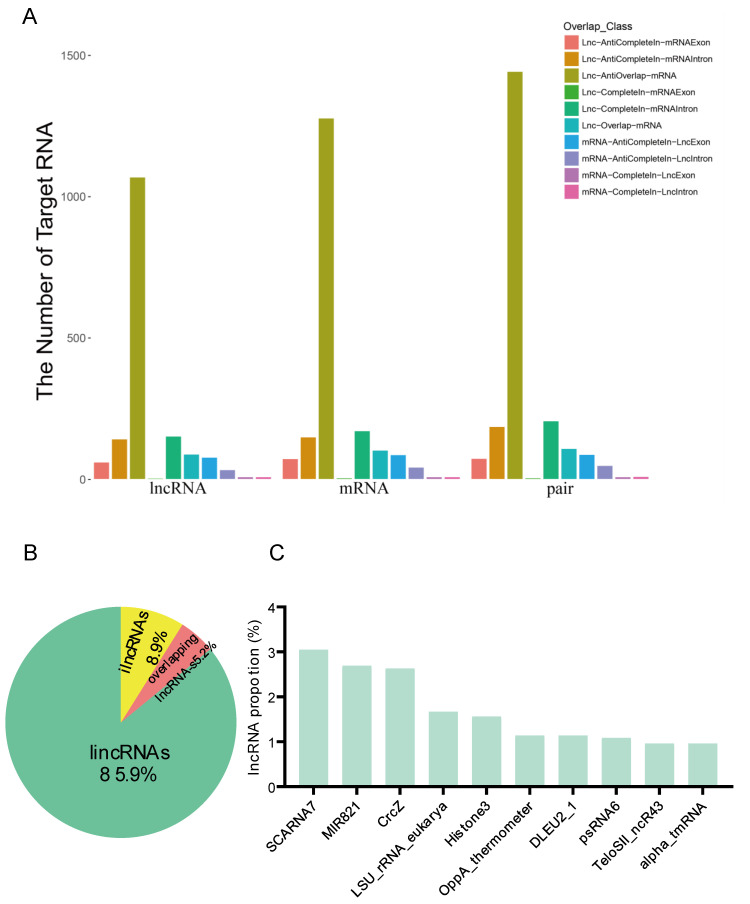
Statistical histogram of the overlapping classifications between the lncRNAs and the mRNA targets (**A**). (**B**) The proportional distribution of the lncRNAs in genomic location and context, where lincRNAs indicates intergenic lncRNAs (85.9%), ilncRNAs indicates intronic lncRNAs (8.9%), and overlapping lncRNAs include sense and antisense lncRNAs (5.2%). (**C**) The top 10 families of lncRNAs in *T. castaneum*, where the X-axis stands for the different families of lncRNAs.

**Figure 5 insects-13-00283-f005:**
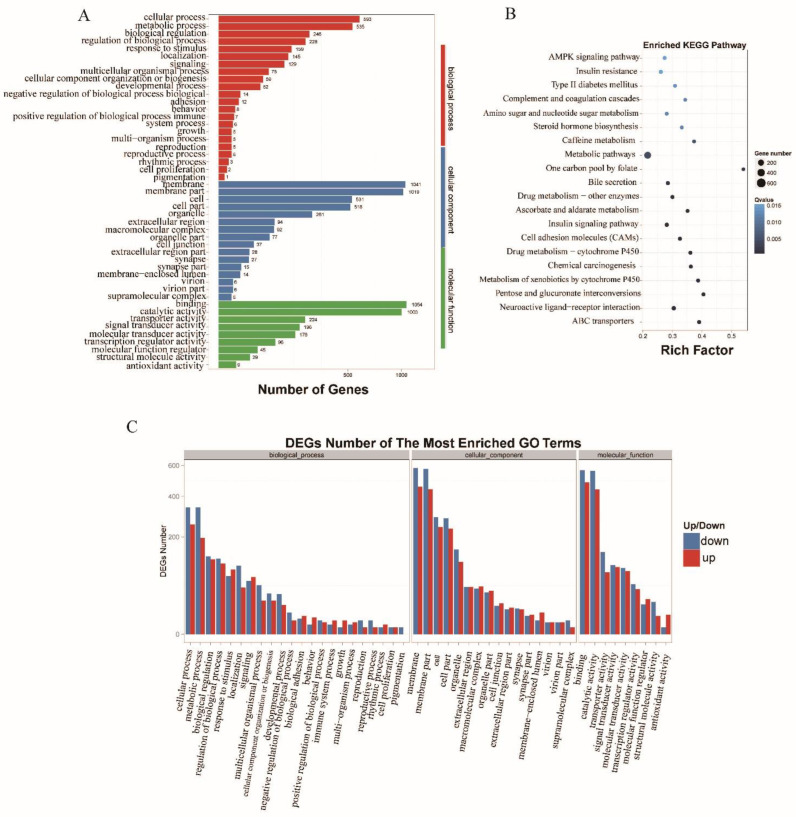
Statistical results for GO enrichment, the Y-axis corresponds to the GO terms, and the red histogram refers to biological process, blue is cellular components, and green is molecular functions (**A**); KEGG pathways (**B**); and up- and down-regulated GO term, where the X-axis indicates the different GO terms, down-regulation is shown in blue, and up-regulation is shown in red (**C**), of significant differential lncRNAs in *Tribolium castaneum.*

**Figure 6 insects-13-00283-f006:**
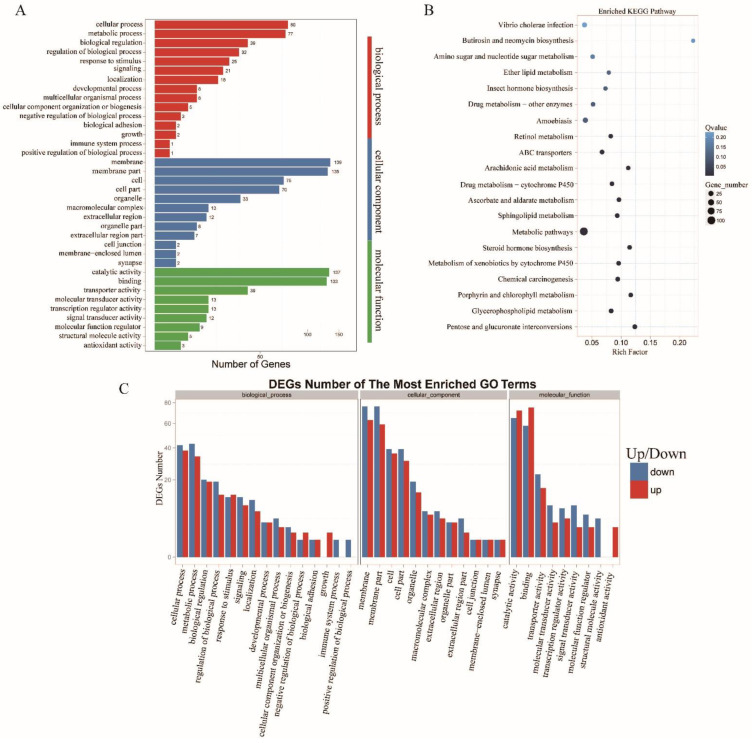
Statistical results from the GO enrichment, the Y-axis corresponds to the GO terms, the red histogram refers to biological processes, the blue is the cellular components, and green is molecular functions (**A**); KEGG pathways (**B**); and up- and down-regulated GO terms, where the X-axis indicates the different GO terms, down-regulation is shown in blue, and up-regulation is shown in red (**C**) of significant differential target mRNAs in *Tribolium castaneum.*

**Table 1 insects-13-00283-t001:** Primers of lncRNAs used for RT-qPCR in *Tribolium castaneum*.

Specific Primer Names	Sequence (5′-3′)	Amplification Size (bp)
RPL18-F	CGACCAAAGGATATGGGATG	198
RPL18-R	GGACCAAAATGTTTCACTGCT	
qLTCONS_00036602-F	GTTCGGACATTTGGTTCAC	80
qLTCONS_00036602-R	AGGCGTTCAGGCATAATC	
qLTCONS_00034604-F	CACCATAGGACTCCAGTT	111
qLTCONS_00034604-R	CAGGTAGGTCAGTTGTCA	
qXR_001574547.1-F	GGTCTTGAAGTGTCTTGATG	81
qXR_001574547.1-R	TGAATATAACGGCGGAGAG	
qXR_511523.2-F	CAAAGGCGGAGAGTTTATG	96
qXR_511523.2-R	TAAGCGACTGTGGGAAATC	
qLTCONS_00020892-F	GGGTCAAGACTCACTTTTG	83
qLTCONS_00020892-R	GTGTCAGTGTCCTAACCT	
LTCONS_00034908-F	GTCTGTCATTCCTTCCAT	96
LTCONS_00034908-R	CTTACCCGTTTCACTTTC	
qLTCONS_00037948-F	GCCTGGAAAGAACAAGAAG	88
qLTCONS_00037948-R	TACTCTCACCTCATCTCACT	
qLTCONS_00035680-F	GTTTGAGGGCAGTAATGTC	133
qLTCONS_00035680-R	TTCGGTAGTCTTCCTTGTC	
qLTCONS_00036125-F	GACCTGTCCTGTTGATTC	101
qLTCONS_00036125-R	CAGCATCTCCTCTTTCAC	
qXR_511525.2-F	CGTTCCGAATGTATGATGAC	96
qXR_511525.2-R	GGCTGCGATGAGATAGTT	
qXR_001575669.1-F	TACGACAGCATCATCTACAG	87
qXR_001575669.1-R	CACGGCGATATTCCTTGA	

**Table 2 insects-13-00283-t002:** Statistics of read align to reference genome.

Sample	Total Clean Reads	Total Mapping Ratio	Uniquely Mapping Ratio
Control_1	79,106,542	75.93%	74.14%
Control_2	97,915,032	71.73%	70.04%
Control_3	76,002,752	74.04%	72.29%
Treated_1	82,392,022	71.54%	69.97%
Treated_2	105,768,982	67.30%	65.79%
Treated_3	75,888,294	72.29%	70.60%

**Table 3 insects-13-00283-t003:** Statistics of all expressed transcripts from the six genome libraries.

Sample	Novel_lncRNA Transcripts	Novel_mRNA Transcripts	Known_lncRNA Transcripts	Known_mRNA Transcripts
Control_1	7329	2648	862	12,346
Control_2	7215	2621	832	12,308
Control_3	7281	2622	843	12,318
Treated_1	7132	2599	804	12,242
Treated_2	7065	2605	811	12,266
Treated_3	7194	2611	827	12,322

## Data Availability

All of the raw sequence data were deposited in the NCBI Sequence Read Archive (SRA) under BioProject accession number PRJNA808823.

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
