# Peer review of "Genome-Wide Identification of the Long Noncoding RNAs of Tribolium castaneum in Response to Terpinen-4-ol Fumigation"

_insects, 2022, doi:10.3390/insects13030283_

Round 1

Reviewer 1 Report

The paper has been improved significantly from previous version. There are still some minor issues that need to be addressed. I am attaching a PDF file with my comments in the specific places. 

Author Response

Dear Reviewer,

Thank you for your useful comments and suggestions.

We have revised the manuscript in response to your advice, and the following shows the point by point response to the your comments. Additionally, We made some other revisions. All the changes in the manuscript were highlighted in RED.

Best Wishes

Yours sincerely,

Min Liao

23-Feb-2022

Comments:

The paper has been improved significantly from previous version. There are still some minor issues that need to be addressed. I am attaching a PDF file with my comments in the specific places.

  1. Line 50-53 need to be rewritten.

Response: Thank you for this suggestion. We have rewrote this sentence.

  1. Line 83-84 need to be rewritten.

Response: This sentence has been rewritten.

  1. Line 217-222 repeats to the M&M section.

Response: We regret for this mistake. We have deleted the duplicate section, and the description has been modified.

In addition, we have corrected other mistakes in this manuscript according to your comment.

Thanks to reviewer’s advice, it’s very carefully.

Reviewer 2 Report

The manuscript “Genome-wide identification of the long non-coding RNAs of Tribolium castaneum in response to terpinen-4-ol fumigation” that assembled the RNA transcripts of T. castaneum before and after terpinen-4-ol treatment and identified the lncRNAs related to mRNA by reducing terpinen-4-ol activity is an interesting study. The manuscript is well written and the authors have rightly analyzed the results. The main concern to me is where the data were deposited. However, I found minor deficiencies that I am enlisting below for their follow up

  1. The qPCR results were analyzed by Livak, K.J.; Schmittgen as mentioned in the materials and methods section. I guess the authors have plotted the results in the form of Figure 3D. The details provided at Line 204-208 are not enough.
  2. Regarding qPCR, please apply statistical analysis so that readers could see the significant differences among the selected lncRNAs
  3. In the results section, please provide statistical analysis details for qPCR results.
  4. Please provide SE for qPCR data.
  5. Figure 3, please provide a Horizontal axis heading.
  6. Why table 1 and figure 3D lncRNAs numbers are different. Please clarify otherwise provide results for all ncRNAs mentioned in table 1.
  7. Figure 5A, 6A are Hard to read please provide better images that could be seen easily.
  8. Please provide a horizontal title for each graph. I do not find it in most cases.
  9. Figure legends are not clear, please provide more details for a broad readership.
  10. Provide the necessary details in the manuscript in section 3.1 about where the data were deposited, it is the most important concern to me.
  11. Discussion is a bit weak, I would suggest expanding by providing a logical discussion of each studied aspect instead of general discussion.
  12. The conclusion must be rewritten with a clear message and wayward from future perspective studies. Please avoid citing any figures in the conclusion.
  13. Figure S1 shows the flowsheet diagram of your scheme of analysis, please also provide here data deposited details such as library accession, etc.

Author Response

Dear Reviewer,

Thank you for your useful comments and suggestions.

We have revised the manuscript in response to your advice, and the following shows the point by point response to the your comments. Additionally, We made some other revisions. All the changes in the manuscript were highlighted in RED.

Best Wishes

Yours sincerely,

Min Liao

23-Feb-2022

Comments:

The manuscript “Genome-wide identification of the long non-coding RNAs of Tribolium castaneum in response to terpinen-4-ol fumigation” that assembled the RNA transcripts of T. castaneum before and after terpinen-4-ol treatment and identified the lncRNAs related to mRNA by reducing terpinen-4-ol activity is an interesting study. The manuscript is well written and the authors have rightly analyzed the results. The main concern to me is where the data were deposited. However, I found minor deficiencies that I am enlisting below for their follow up.

  1. The qPCR results were analyzed by Livak, K.J.; Schmittgen as mentioned in the materials and methods section. I guess the authors have plotted the results in the form of Figure 3D. The details provided at Line 204-208 are not enough.

Response: We agree with you. We have provided necessary information in the revised manuscript.

  1. Regarding qPCR, please apply statistical analysis so that readers could see the significant differences among the selected lncRNAs.

Response: Thank you for your advice. We have provided necessary statistical analysis details for qPCR. 

  1. In the results section, please provide statistical analysis details for qPCR results.

Response: Done.

  1. Please provide SE for qPCR data.

Response: Done.

  1. Figure 3, please provide a Horizontal axis heading.

Response: We agree with you. The horizontal axis heading has been provided in the revised manuscript.

  1. Why table 1 and figure 3D lncRNAs numbers are different. Please clarify otherwise provide results for all ncRNAs mentioned in table 1.

Response: We regret for this misunderstanding due to the primers were listed at the end of the Table 1 in the submited manuscript. Actually, the data is the same, we have adjusted the table in the revised manuscript to avoid the misunderstanding.

  1. Figure 5A, 6A are Hard to read please provide better images that could be seen easily.

Response: We regret for this mistake. We have replaced this figures with the better ones.

  1. Please provide a horizontal title for each graph. I do not find it in most cases.

Response: Done.

  1. Figure legends are not clear, please provide more details for a broad readership.

Response: Thank you for this suggestion. We have modified the figure legends.

  1. Provide the necessary details in the manuscript in section 3.1 about where the data were deposited, it is the most important concern to me.

Response: We agree with you. We have built a BioProject in NCBI, and the accession ID is PRJNA808823, however, we have written an email requesting NCBI staff to correct it into Multiisolate due to we chose the wrong Scope. The data hasn't been uploaded so far because of the content of 50 Gb. Biosamples and SRA accession would be linked to this BioProject in the next few days.

  1. Discussion is a bit weak, I would suggest expanding by providing a logical discussion of each studied aspect instead of general discussion.

Response: Thank you for this suggestion. We have modified the discussion section according to your suggestion. This description is more concise in the revised manuscript.

  1. The conclusion must be rewritten with a clear message and wayward from future perspective studies. Please avoid citing any figures in the conclusion.

Response: Thank you for this suggestion. We have rewrote the conclusion according to your suggestion.

  1. Figure S1 shows the flowsheet diagram of your scheme of analysis, please also provide here data deposited details such as library accession, etc.

Response: Thank you for this advice. We have provided necessary information.